# Understanding Teacher Resilience: Keys to Well-Being and Performance in Chilean Elementary Education

**DOI:** 10.3390/bs15030292

**Published:** 2025-03-02

**Authors:** Sonia Salvo-Garrido, Pilar Cisternas-Salcedo, Karina Polanco-Levicán

**Affiliations:** 1Departamento de Matemática y Estadística, Universidad de La Frontera, Temuco 4780000, Chile; 2Independent Researcher, Temuco 4780000, Chile; cisternas.pilar@gmail.com; 3Programa de Doctorado en Ciencias Sociales, Universidad de La Frontera, Temuco 4780000, Chile; k.polanco01@ufromail.cl

**Keywords:** resilience, well-being, self-care, teacher, elementary education

## Abstract

Teachers’ ability to manage stress and daily challenges is crucial to their professional development and well-being. By examining how Chilean elementary school teachers perceive the growth in their resilient behavior, this study seeks to determine the predominant factors that foster resilience in these individuals. Sixty-three public school teachers participated in semi-structured interviews as part of a qualitative analysis employing deductive and inductive coding. The results show that teacher resilience is built through a complex interaction between personal, professional, and contextual factors. Among the most important factors are self-care, psychosocial support, and an institutional environment that facilitates teachers’ emotional well-being. Relationships with students and their families also support this resilience, creating an atmosphere of trust and collaboration in the classroom. The study highlights that strengthening resilience not only improves the teaching experience but also has a positive impact on students’ well-being and academic performance. Integrating self-care strategies, support networks, and family alliances in teacher training and education policies is essential to developing healthy and sustainable school environments. The study concludes by emphasizing the importance of specific training programs that give teachers effective tools to manage stress and adversity better. Although limited to elementary education in Chile, this study invites future research that expands the analysis to other education levels and cultural settings, offering a more comprehensive view of resilience in education.

## 1. Introduction

One of the most intricate aspects of the social structure and one of the cornerstones in the construction of society is the teaching profession ([77]; [94]). In this light, teacher resilience is essential for the professional performance of elementary school teachers, as it empowers them to confront, manage, and resolve challenging situations ([28]; [97]). Teachers convey knowledge and shape future generations’ character, social skills, and critical thinking ([39]; [74]; [75]). However, daily challenges arise, such as teaching in an inclusive and multicultural educational environment, managing their students’ emotions so they can learn, adapting to administrative changes, and balancing their professional and personal lives ([20]; [43]; [70]; [103]). As societies become more diverse, teachers must navigate an increasingly complex environment, requiring that they not only develop skills to teach their students but also adaptability and resilience ([5]; [8]; [44]; [76]). In this context, understanding the complexities of the teaching profession is essential to adequately supporting and valuing the role of teachers in society ([9]; [18]; [98]).

Resilience is understood as the ability to adapt and recover in the face of adverse situations ([59]; [78]; [80]); it develops progressively and varies depending on different personal, social, and contextual factors ([37]). Resilience has become a fundamental educational construct, especially in teacher training and practice. Cultivating resilient behavior in teachers is crucial to overcoming the challenges inherent in the profession ([27]; [37]; [104]) and to encouraging the development of children’s resilience. School is a unique environment where students can acquire academic skills and develop their own resilience ([56]). In a context where teachers face constant challenges, from managing diverse classrooms to administrative and emotional demands ([92]), resilience becomes a critical element for well-being and professional effectiveness ([48]; [97]). In this sense, the correlation between resilience and well-being in teachers is substantiated by several studies ([46]; [72]; [88]; [99]). Specifically, teachers’ resilience aids in the development and preservation of their sense of well-being ([38]), as it also promotes the advancement of their students ([99]).

Resilience is a process that leverages the resources available at different levels, including personal (individual traits), social (support networks), and contextual (institutional environment and education policies), enabling the effective management of situations and the resolution of difficulties in the best possible way. Hence, it is a fundamental aspect of both teacher training and professional practice ([35]; [56]). On a personal level, it is important to mention that resilience is related to socioemotional competencies, considering that greater self-awareness and emotional regulation can be observed; specifically, teachers with less developed resilience present symptoms of stress ([2]; [15]). In addition, they experience burnout and difficulties maintaining discipline in the classroom ([28]; [58]; [73]; [79]; [81]). They may also have depression and anxiety ([48]; [104]). In contrast, those teachers with greater resilience show greater self-awareness, self-regulation, autonomy, empathy, collaboration, and a better mood ([2]; [48]; [104]), which may lead to self-care practices ([52]; [69]; [101]; [104]). Such teachers exhibit heightened awareness of the issue, which will positively impact the resolution of difficulties, enhancing their perception of resilience ([29]). [4] ([4]) report gender differences in resilience-related coping styles, noting that men focus on rational elements, whereas women use emotion-focused strategies to solve problems. Notably, the greater development of socioemotional competencies associated with teacher resilience promotes students’ emotional regulation as they employ more effective coping mechanisms to manage daily challenges ([39]). This improves student learning ([34]; [42]).

On the contextual level, resilience is related to occupational well-being ([17]; [19]), the opportunities for professional development ([44]), and academic training ([23]; [51]; [94]; [100]). This ensures that the teacher is equipped to confront issues in the classroom and the teaching–learning process within a dynamic framework ([44]; [54]). In this sense, the teacher’s experience is relevant given that it is associated with greater resilience ([50]; [53]; [57]; [100]) even though there are contrasting results ([23]; [100]).

Furthermore, peer and school leadership social support is important for dealing appropriately with adversity as it is a resource that can be used to resolve difficulties better ([3]; [17]; [40]; [41]; [95]). Thus, teachers must be encouraged to provide emotional support and collaborate, accompanying those with less experience ([30]). This aids in managing conflicts in the classroom ([58]) and effective school practices, improving teachers’ perception in the professional setting ([48]; [68]).

In the same vein, on the social level, in addition to the support of colleagues and school administrators, the teachers’ families are another significant support network. There is evidence that maintaining a work-life balance promotes resilience. The family is a fundamental support network for teachers, protecting and strengthening their resilience ([30]; [89]). It is primarily beneficial for teachers facing personal difficulties; thus, it would enhance the practice of the profession ([13]). It should be noted that support networks may vary in size and composition over time ([11]).

Importantly, students’ families can positively or negatively influence teacher resilience ([31]; [32]). Teachers recognize the relevance of establishing appropriate relationships with students’ families to enhance their engagement in education activities and promote mutual contributions; for example, parental involvement in school tasks yields positive outcomes for learning ([24]; [60]; [87]; [90]). However, teachers must fulfill different responsibilities and activities that occasionally obstruct this objective ([24]). Parental involvement improves teaching commitment, and the collaboration between parents and teachers reduces parental burnout when involved in their children’s education ([55]). Therefore, teachers with greater relational competencies address challenges more effectively, thereby preserving their commitment ([105]). It is relevant that there be a shared school–family project ([55]); however, it is often perceived that greater collaboration from families is needed ([6]).

It is important to note that Chile has a segregated education system characterized by reduced segregation among students from higher socioeconomic strata, while segregation intensifies among the lower strata ([62]; [61]). This adversely affects teachers, as they must teach and meet objectives at schools predominantly attended by disadvantaged students ([63]). Accordingly, in challenging contexts and with students in vulnerable conditions, the students’ academic performance may suffer ([33]). Consequently, resilience cultivates strong bonds within a community, enabling its members to confront infrastructural restrictions, isolation, and the vulnerabilities faced by students and their families ([86]).

Concerning resilience among Chilean elementary school teachers, [97] ([97]) assert that resilience is a protective factor against stress and mental health issues, enhancing the perception of well-being. Self-efficacy and resilience significantly impact teachers’ prosocial conduct, which is crucial for cultivating positive relationships between teachers and students ([83]). A qualitative study that included elementary and secondary education teachers indicates that resilience at the individual level is related to the development of teaching–learning skills. In addition, at the relational level, it is linked to leadership and constructive relationships among various educational community members. At the institutional level, resilience is fostered through professional autonomy and the assistance provided by experts in different fields that enable teachers to do their work ([96]). Moreover, enhanced resilience is noted in women and older adults ([14]), and it is concurrently found that teaching experience correlates positively with resilient behavior ([97]).

Research in the field of teacher resilience has gained relevance in recent decades, driven by the need to understand not only the individual characteristics of teachers ([12]) but also the impact of the school and social environment on their ability to overcome difficulties ([1]). A qualitative approach is used to explore the teachers’ narratives, enabling a comprehensive understanding of their subjective and intersubjective experiences ([26]) and the identification of patterns that promote resilience, thus facilitating an in-depth understanding of the phenomenon. It is also relevant to consider the implications of teacher resilience not only for teachers but also for their students and the educational community as a whole ([36]; [82]; [93]). A resilient teacher can manage their own stress and adversity and act as a role model for their students, fostering a positive and motivating learning environment ([32]; [71]). In addition, their resilience influences decision-making and transforms coexistence within the educational community ([84]; [91]). Understanding the factors that promote resilience in teachers is fundamental to ensuring their well-being, improving teaching quality, and promoting students’ comprehensive development ([85]; [80]).

Building on this foundation, this analysis aims to contribute to understanding resilience as a dynamic process ([27]; [104]), which is nurtured by the interaction between the individual and their environment. This is essential for the sustainability of the teaching profession in an ever-changing world. Therefore, the idea is to contribute by providing greater clarity on how teachers can cope with high levels of stress, work overload, and conflict situations, using protective factors to maintain their motivation and commitment to teaching ([7]; [28]). Identifying the factors that foster teacher resilience has relevant implications for initial and ongoing teacher training and for developing programs that strengthen these skills and competencies in teachers ([5]; [15]; [16]; [28]; [64]), better preparing them to face professional challenges and preventing talent drain ([22]). Similarly, executing measures that cultivate resilient work environments enhances teachers’ well-being and professional efficacy throughout their careers.

Indeed, understanding teacher resilience is essential to ensuring teacher well-being and effectiveness, as well as to improving teaching quality and fostering comprehensive student development. Therefore, this study aims to identify the predominant factors that promote resilience in elementary school teachers in Chile by qualitatively analyzing their perceptions of the development of their resilient behavior.

## 2. Methods

### 2.1. Participants

The study sample consisted of 63 elementary school teachers, 73% employed in urban areas and 27% in rural settings. Regarding institutional affiliation, 50.8% belong to the Local Public Education Service (SLEP), whereas 17.5% work in the Municipal system. Table 1 indicates that most participants are women (82.5%), with a mean age of 40.2 years (SD = 12.3, range: 24–70). Regarding academic qualifications, 14.3% hold a master’s degree, 1.6% a doctorate, and 27.0% have completed a post-degree specialization. Participants also have diverse teaching experience, with 30% having 1–5 years, 45% having 6–15 years, and 25% more than 15 years (Table 1). This distribution facilitates an analysis of resilience across different stages of professional development.

### 2.2. Data Collection Techniques

The Chilean educational system presents unique challenges for elementary school teachers. Structural inequalities, demanding workloads, and constant educational reforms shape their professional experiences. Public and subsidized schools often face resource limitations, and teachers frequently work under high-stress conditions due to standardized testing pressures and student diversity. Additionally, socioeconomic disparities influence teaching environments, requiring educators to develop adaptive strategies to navigate these challenges. Given this context, understanding teacher resilience in Chile requires considering both personal coping mechanisms and systemic factors affecting their professional well-being.

A semi-structured interview framework was created to explore teacher resilience. The interview protocol was founded on existing theoretical frameworks and validated by a panel of experts in education and psychology to guarantee clarity, coherence, and relevance. A pilot test was conducted with a small cohort of teachers to refine question phrasing and assess the instrument’s effectiveness.

The interview guide was designed as a versatile instrument to allow participants to elaborate on their experiences and reflect on their approaches to challenges in their teaching practice. The questions focused on key areas such as self-assessment of resilience, career transitions, and strategies for surmounting difficulties in the school setting. The interview also examined personal, social, professional, and contextual variables that teachers deemed essential for fostering resilience, such as adaptability, persistence, and social support.

To ensure data collection reliability, all interviews were audio-recorded and transcribed verbatim, preserving accuracy for qualitative analysis. Participants were assured of confidentiality, and informed consent was obtained prior to the interviews. The study adhered to the ethical standards established by the university’s research ethics committee.

Given the nature of qualitative analysis, a systematic and iterative approach was employed to improve the reliability of the findings ([65]). A single-researcher qualitative content analysis ensured a uniform analytical perspective throughout the process. Although collaborative analysis is often beneficial, single-coder analysis can yield consistent and high-quality results when applied with a systematic and rigorous methodology. This approach mitigates potential inconsistencies arising from multiple coders while enabling a cohesive and in-depth interpretation of the data ([45]). Peer debriefing and external audit methodologies were also incorporated to augment credibility and ensure analytical accuracy.

### 2.3. Procedures

The recruitment of participants and the execution of fieldwork adhered to a systematic method to ensure compliance with ethical and institutional requirements. Formal communication was established with school administrative teams to secure the necessary authorizations for conducting the study. Once approval was granted, prospective participants were apprised of the research objectives, the procedures, and their rights as participants.

To uphold ethical principles, informed consent was obtained from all participants before data collection, ensuring voluntary participation and confidentiality. The study adhered to the ethical standards set by the university’s ethics committee, and the research protocol received formal approval by the Scientific Ethics Committee of the Universidad de La Frontera (Assessment Report N°053_21; Study Protocol Page N°019/21).

### 2.4. Data Analysis

This study employed a qualitative approach with an exploratory design to identify factors influencing teacher resilience. Interviews were transcribed and analyzed using ATLAS.ti 25, applying relational content analysis to establish connections between emerging categories and concepts, facilitating a deeper understanding of how contextual and personal factors shape resilience ([49]).

The analysis followed three phases. First, a deductive phase structured the data within a predefined theoretical framework. Next, an inductive phase applied open and in vivo coding to identify emerging patterns ([10]). Finally, in the explanatory phase, the systematic reorganization of codes uncovered underlying relationships and meaningful connections within the data ([47]).

To evaluate the relevance of emerging codes, this study used the G value as a saturation metric. In ATLAS.ti, the G value (Groundedness) represents the frequency with which a code is linked to citations, indicating its empirical support. This metric serves as an epistemological foundation in qualitative methodology, reflecting the iterative process of data saturation ([25]). Data saturation occurs when no new information emerges, reinforcing the reliability of the findings. The process involved systematically assigning codes to relevant text segments, quantifying citations per code, and establishing a threshold of G ≥ 30 for inclusion in visual analyses. Codes below this threshold were excluded to prevent information overload.

The rigorous selection of codes ensures traceability to the original data, enhancing the validity of interpretations through constant comparison of contexts and coding processes ([21]). This methodological approach strengthens the credibility of findings by refining interpretations based on contextualized data analysis.

Despite variations in individual narratives, no consistent patterns linked resilience strategies to years of teaching experience. This suggests that resilience development is influenced by an interplay of personal, social, and contextual factors rather than professional trajectory alone.

## 3. Results

The configuration of resilient behavior in teachers is a multidimensional process (Table 2) that synergistically articulates both individual and organizational elements in the educational context. In this respect, teachers rely on different elements to overcome the challenges inherent to the teaching profession. They identify two contexts where they demonstrate factors that promote the development of resilient behavior: resilience in their personal life and the professional environment.

For ease of understanding, the codes are hierarchically structured as Dimension and Codes (the latter indicated in italics).

### 3.1. Development of Resilient Behavior: From the Personal Level

The development of resilience from the personal level encompasses 30 codes (Table 3), categorized into four dimensions: Personal (21), Family (3), External Factors (4), and Organizational (4), described in Table 2. Table 3 delineates teachers’ multifaceted development of resilience, spanning their personal environment, interaction with students’ families, relationships with contextual factors, and connections with the management team from an organizational perspective.

The generation of visual representations was guided by the hierarchical organization of codes and the G value (≥30). Additionally, lower-frequency codes with strong theoretical significance were considered. To enhance readability, overlapping codes were excluded.

The visualization process followed a structured sequence: first, codes were categorized into predefined dimensions. Then, ATLAS.ti analyzed conceptual proximity between codes to identify interconnections. Figures were arranged progressively—starting with central themes, followed by relationships between dimensions, and concluding with detailed interactions—ensuring systematic and logical presentation of information.

For all figures, Solid lines represent the interaction and interdependence between dimensions, while dotted lines indicate the relationships between the corresponding codes.

Figure 1 schematically illustrates the characteristics that a teacher considers useful for developing resilience in the personal context.

From Grounded Theory, “groundedness” (G) represents the connection between theory and data obtained through constant comparison, measuring the connection frequency between a code and direct quotes, quantifying the quotes associated with each code, and indicating the level of empirical grounding of the theoretical constructs.

According to Figure 1, the development of resilient behavior from a teacher’s personal perspective is significantly influenced by the ability to manage emotions, evidenced by its high saturation (G = 108) and multiple connections with other system components. Emotional management acts as an articulating axis that impacts several areas, from the support network (G = 59) to confidence and commitment (G = 45), through the management of personal problems (G = 42) and the ability to ask for help (G = 38). This interrelationship suggests that a teacher with strong emotional management skills is likelier to develop effective, resilient behavior.

On a personal level, the teacher identifies five key characteristics that contribute to developing resilient behavior: emotional management, management of personal problems, ability to ask for help, ability to self-analyze, and self-care in health. By being self-aware of their real skills, the teacher who reflects on their strengths and weaknesses recognizes their vulnerability, assuming they are incapable of the full load and willing to depend on someone else. In this sense, the skills related to managing their problems are the most significant; a teacher who remains emotionally and mentally healthy, who is reflective and humble, is more able to develop resilient behavior.

Reflection: *“… because many times you make mistakes and realizing that you made a mistake helps you not to repeat that behavior.”*(Interview M15).

Asking for Help: *“… also the ability to accept when you’re wrong and when you have to say yes, you have to look for another alternative and suddenly also to ask for help because I think that one of the ways teachers fall short is not being able to ask for help because they feel they are self-sufficient…”*(Interview P2).

Emotional Management: *“…not to be overwhelmed by the situation, because on many occasions students get frustrated and obviously you have to stay calm and give them the confidence that they can continue.”*(Interview M11).

Management of Personal Conflicts: *“… and the pitfalls that arise along the way, whether in interpersonal relationships, with children, with lawyers, with colleagues, etc., are pitfalls that you have to overcome, know how to deal with…”*(Interview S3).

Health: *“… but if I’m not well, it’s impossible to control these situations…”*(Interview V12).

At the family level, communication between the teacher and the student’s family is fundamental for creating a joint strategy, where the student’s family’s trust, support, and commitment are pillars for developing resilience in the teacher.

Communication and Dialogue with Families: *“… I would make an appointment with the parent and connect with the family…”*(Interview V2).

Family Confidence and Support: *“… parents need to focus on being able to trust you…”*(Interview V21).

At the External Level, the teacher develops resilience by building links between all actors in the educational community and those external to it; effective collaboration must be promoted, and the educational process must be reinforced. Building strong and meaningful relationships between teachers, students, families, and the wider community contributes to a more inclusive and enriching learning environment. These ties not only facilitate communication and the exchange of resources but also promote a sense of belonging and mutual support, essential elements for the integral development of students and the well-being of the entire educational community.

As described above, teachers who recognize their weaknesses can rely on mental health professionals to work on their emotions.


*“… what influences me the most, what I feel has influenced me the most is the issue of support networks […] in other words, I feel that the more support networks I have, my resilience increases…”*
(Interview P3).

At the organizational level, the teacher emphasizes the importance of the quality of relationships within the school. In this study, the teachers recognize that a good work environment directly affects the development of personal and professional resilience.


*“… that you like where you’re working and feel valued and well-treated within the field, work environment; I think that’s very important, but for me the most important thing is that you like it, that you want to be there, that you arrive, get out of the car, the bus, the taxi when you arrive, see the door of your school and enter with enthusiasm, empowered, you get there early, fresh and with an idea, and you confront situations and offer solutions”*
(Interview S3).

### 3.2. Development of Resilient Behavior: From the Professional Level

The development of resilient behavior consists of 29 codes, described in Table 1, distributed in four areas: Teacher, Family, External, and Organizational Factors (Table 4).

Figure 2 illustrates that the teacher identifies classroom conflict management as the main attribute affecting all the dimensions specified. This study identifies proper conflict management as promoting a sense of professional satisfaction in teachers. This emotional well-being encourages the development of resilience from a professional perspective.

At the professional level, the analysis reveals five fundamental dimensions that contribute to the development of teacher resilience: school conflict management (G = 174), adaptability (G = 163), teaching experience (G = 148), empathy (G = 137), and motivation (G = 100). The teacher who can adjust to changing situations and adopt new teaching approaches according to the needs and context (G = 33) demonstrates the importance of the environment developing resilient strategies. This ability to adapt influences the creation of strategies to deal with challenging situations, where experience (G = 148) contributes with a repertoire of successful strategies. Motivation and the ability to create emotional connections (G = 137) with their students and peers play a relevant role in building teacher resilience, particularly when linked to institutional support and commitment (G = 45), which allows them to maintain a positive attitude even in difficult times.

Consequently, these interconnected links suggest that teacher resilience at the professional level is founded on a multidimensional framework where conflict management and adaptability to new situations serve as the fundamental pillars bolstered by accrued experience and socioemotional competencies.

Adaptation: *“… when something unforeseen happens, no, I don’t get frustrated; on the contrary, I know how to cope well in those situations…”*(Interview M16).

Conflict Management: *“… I think that the problems students bring from home, perhaps crises they face daily and situations that may arise momentarily in the classroom…”*(Interview V1).

Work Experience: *“… it has changed my behavior in how I deal with things, the tools I have, because clearly the experience gives you other tools and has changed my interaction with the children in general because that’s when you realize how each person reacts. You also realize that there are things that work with some children that don’t work with others, so basically, the experience has made this change…”*(Interviews M12).

Motivation: *“Motivation. The motivation to want to be there, to want to help, to want to improve, and to want to continually educate the kids, to give the best of yourself.”*(Interview M10).

Empathy: *“… seeing the student thinking that he’s my child, for example, that you may be facing this situation and obviously the look changes, because you would like your child to be treated as you treat other people’s children…”*(Interview M8).

At the Family Level, teachers’ support and commitment (G = 45) from their students’ families can influence their emotional and professional well-being. When families are engaged and supportive of the educational process, teachers feel valued and supported, which can increase their motivation and job satisfaction. The family’s support is crucial in formulating successful resilient strategies, particularly when combined with school conflict management skills (G = 174), which emerge as the most extensively documented element in the analysis.

Family support is a catalyst that enhances the teacher’s professional well-being and adaptability, establishing a solid foundation for developing resilient behaviors in the educational context.


*“… the constant support of the family, that they come to school when you call them, that they participate with the school…”*
(Interview V6).

With respect to external factors, teachers operating in settings characterized by substantial economic and social vulnerability (G = 33) have considerable challenges that may lead to stress and emotional exhaustion. However, resilience develops when teachers find ways to adapt to these adversities. The teacher’s ability to adapt to contextual challenges develops in response to the specific demands of the sociocultural environment, enhanced by institutional support and commitment (G = 45), teaching experience (G = 148), and empathy (G = 137), acting as modulating factors in the relationship between contextual challenges and the development of resilient strategies. This finding suggests that the capacity to respond to environmental adversities is built by integrating professional competencies and socioemotional skills, especially when these are articulated with the support networks available in the educational context.


*“… many times, I went to the student’s house, and there I realized the environment in which the student lived, where 6 adults slept in one room and the child, so there was no protection of privacy, nothing, everything that implies, the overcrowding in a home…”*
(Interview V2).

The school’s support network (G = 59) stands out at the organizational level, comprised of professionals in charge of school coexistence, school integration programs, staff, and management. This intra-school relationship is critical to understanding how collaborative contexts within a school can influence teachers’ ability to adapt and thrive in the face of challenges. These interactions can be a valuable resource for developing teacher resilience, as they constitute a fundamental component in building adaptive and sustainable educational communities, fostering a sense of belonging, mutual support, and collaboration.


*“… I feel that the more support networks I have, or the bigger [the support network], the better the support; I feel that my resilience increases…”*
(Interview P3).

### 3.3. Development of Resilient Behavior: Students and Their Families

The relationship between students and their families and the development of resilience in teachers is a crucial aspect that influences education dynamics and teachers’ well-being. When families are engaged and actively support the educational process, teachers tend to experience a more positive and collaborative environment. This family support can manifest in various ways, such as participation in school activities, monthly meetings, and willingness to collaborate with teachers in problem-solving. Such interaction strengthens the bond between school and home and provides teachers with a sense of support and recognition, which is fundamental to their resilience.

As shown in Figure 3, the student’s ability to manage their emotions is the most binding characteristic for the development of resilience in the teacher, causing the relationship between families, the teacher, and the educational community to create a safe environment.

From the student level, three characteristics influence the development of resilient teacher behavior: emotional management (G = 73), trust and respect for the teacher (G = 81), and the creation of a pleasant, safe, and pedagogically stimulating classroom environment (G = 72). When students learn to manage their own emotions, this is reflected in the classroom climate, reducing the incidence of disruptive behavior. This student behavior directly affects the teacher’s mental health and emotional management so the teacher can focus on teaching and developing meaningful relationships. This, in turn, creates a more conducive environment for teachers to feel supported and motivated, which strengthens their resilience. When students trust their teacher, they are more willing to participate actively in the learning process and face academic challenges. This trusting relationship improves classroom dynamics and makes teachers feel valued and effective in their work. Confidence fosters an environment where teachers can experience less anxiety and stress, contributing to their ability to adapt to adverse situations and maintain a positive attitude.

The integration of these components lays the foundation for sustainable development of resilience, where trust, emotional management, and environmental safety act as catalysts in building adaptive and emotionally healthy educational communities.

Emotional Management: *“… this work has a lot, a lot of that emotional area, I believe that 90% of my students have, either the family or the student, some emotional burnout, then communication is needed, all the emotional stuff is needed, not just the academic…”*(Interview V16).

Trust and Respect for the Teacher: *“… respect for each other, among the parents, the parents for the teacher, the teacher for the students…”*(Interview V3).

Pleasant and Safe Classroom Environment: *“… if a student is comfortable in a room, obviously they are going to create a safe space…”*(Interview S7).

The family-level analysis in developing teacher resilience reveals a significant correlation between communication and dialogue (G = 22) and the structures of support and commitment (G = 45), where communication is a mediator between family dynamics and the development of socioemotional competencies in the school environment. From the student’s point of view, the family is a microsystem where children learn to identify and regulate their emotions, which lays the foundation for their emotional and social development. When parents actively participate in their children’s emotional education, promoting empathy and self-regulation and making communication and dialogue a practice of bonding rather than “questioning,” students tend to develop greater emotional intelligence. This ability allows them to manage their emotions in the classroom better, which reduces the incidence of disruptive behavior and conflict, creating a more positive learning environment; therefore, the teacher develops their resilience more organically.

The integration of family communication and institutional support structures shapes an ecosystem that facilitates the organic development of teacher resilience. This interconnection of relationships suggests that teacher resilience is built by systematically expressing socioemotional competencies, effective communication practices, and family–institution support structures.

Support, Trust, and Commitment: *“… the work with the family, constant feedback…”*(Interview V10).


*“… I am very close to my parents, in fact, the previous year they told me to continue with my third-grade class so as not to leave the children aside…”*
(Interview V22).

Communication and Dialogue: *“… I am a faithful believer that the link between family and school has to be very important and constant…”*(Interviewee M16).

From the organizational level, the internal support networks (G = 59) contribute to creating safe and pleasant environments through the professionals in charge of school coexistence, teaching integration, health personnel, and the educational community in general. School support networks are essential for teachers’ professional and emotional development. These networks make the exchange of experiences, strategies, and resources possible, which helps teachers face the daily challenges of their work. When teachers feel supported by their colleagues, their ability to adapt to adverse situations is strengthened, contributing to the development of resilient behavior. Similarly, when students and their families feel supported by teachers and the educational community, emotional regulation fosters resilience.

The effectiveness of organizational support networks transcends the teaching environment, extending to constructing a resilient educational community that integrates the needs and resources of both students and their families, thus shaping a system to develop resilient competencies at multiple levels.


*“… If my resilience has changed? Also, what influences me the most, what I feel has influenced me the most is the issue of support networks…”*
(Interview P3).

## 4. Discussion

The study’s findings underscore the importance of teacher resilience as a multidimensional construct ([92]), dynamic in nature and influencing emotions, cognitions, and behaviors in addressing daily challenges ([28]; [37]; [59]). Teacher resilience focuses on managing educational challenges ([104]); therefore, it is essential to understand those characteristics that promote teacher resilience. It is important to highlight that the Chilean education system is characterized by significant segregation, necessitating that some teachers engage with students and families in vulnerable situations. Moreover, low family academic expectations and subpar student performance can adversely affect their job performance ([33]; [62]; [61], [63]). This study sought to identify the factors that foster resilient behavior in teachers from a multisystemic perspective, analyzing the interactions among personal, family, professional, and contextual factors and how they manifest in challenging and difficult situations ([1]).

Building on these insights, the findings reveal a multidimensional architecture of resilient teacher development, where emotional management (G = 108) emerges as the central axis in the personal sphere, while in the professional dimension, school conflict management (G = 174) and adaptability (G = 163) stand out as structural components. This configuration is complemented by the student’s trust in the teacher (G = 81) and the student’s emotional management (G = 73), acting as catalysts in the construction of safe school environments, combining with the organizational sphere through the institutional support networks (G = 59), which facilitate the systemic integration of effective, resilient responses.

The evidence documents the inter-relational ([1]) and dynamic nature of the factors promoting resilient development in educational contexts, where each identified dimension contributes synergistically to building sustainable adaptive capacities, aligning with the identification of elements that encourage teacher resilience from a multisystemic perspective.

### 4.1. Factors Promoting Teacher Resilience: From Personal Life

According to the findings of this study, resilience involves teachers’ ability to manage their emotions, empathize, reflect on their experiences, and seek support when needed without considering help-seeking as a sign of weakness or victimization, since healthy relationships promote resilience ([2]; [66]). Consequently, it is evident that resilience functions as a protective factor among Chilean elementary school teachers, as it mitigates stress and improves mental health, thereby increasing overall well-being ([97]).

Building on this understanding, it is emphasized that self-care entails emotional and personal problem management, reflection, and the ability to ask for help ([69]; [101]; [104]), which makes it possible to face and overcome challenges from the personal sphere ([12]). Self-care is one of the main factors in developing resilience as teachers require the time to take care of their mental health ([69]; [101]).

Psychosocial support involves family and external networks supporting the teacher, contributing significantly to emotional and mental health ([30]; [89]). These networks are essential in balancing personal and professional life, helping teachers maintain their well-being and build resilience in the face of external pressure ([13]; [30]). Teachers who receive sufficient psychosocial support are more aware of the usefulness of self-care techniques ([17]; [53]). It is important to emphasize that network interactions are based on communication and dialogue between the parties.

Proactive institutional support groups’ characteristics related to the emotional work environment: pleasant work environment, commitment, support, and trust. Participants in this study underscore the importance of an emotionally healthy work environment where teachers feel supported by their colleagues and the administration, as teacher performance depends on the support of the management team ([30]; [40]; [41]).

A work environment fostering trust, commitment, and general well-being is crucial for developing teacher resilience, as it provides a support network that helps them manage stress and work difficulties ([68]). A study involving Chilean elementary school teachers indicates that resilience is related to prosocial behavior, specifically, the helping and cooperative behaviors that teachers exhibit, which promote positive interactions ([83]; [96]).

### 4.2. Factors Promoting Teacher Resilience: From Professional Life

Access to the resources that teachers obtain to strengthen their professional resilience within the educational community through social interactions is considered relevant ([80]). This concept integrates elements such as administrative organization and the commitment of students’ families and networks pertaining to school coexistence and teaching integration. A strong school network and good organization within the school environment are essential for fostering teacher resilience ([3]; [31]; [88]). It also provides them with a sense of belonging and purpose within the school community, which is critical in times of crisis ([50]), promoting climate stability and reducing teacher turnover ([22]).

Conflict Management is listed as a critical component of teacher resilience, with containment strategies to facilitate and improve problem-solving ([68]). Teachers who develop effective conflict management skills ([4]; [29]; [102]) can maintain a positive learning environment and minimize stress related to difficult interactions, which in turn strengthens their resilience ([2]). In this sense, Chilean teachers regard professional autonomy and the presence of trained professionals to assist in decision-making under challenging situations as beneficial to their resilience ([96]).

Work experience and internal motivation contribute significantly to resilience. Teachers with more experience tend to have more tools and strategies to face challenges, while internal motivation drives them to persevere in the face of difficulties ([50]; [53]; [57]; [100]) and to perform better ([67]). Coincidentally, it has been found that teaching experience is positively correlated with resilient behavior ([97]).

Teacher adaptability is a central theme for resilience, particularly in constantly changing school environments. Adaptability allows teachers to adjust to different social and cultural contexts, as well as to deal with external factors that appear suddenly or unexpectedly that could affect their performance and well-being ([5]; [76]). It is important to consider that Chilean teachers frequently operate in underprivileged environments, where they lack the necessary material resources and face limitations both in infrastructure and in connectivity or access; therefore, resilience and adaptability are important ([86]).

### 4.3. Factors Promoting Teacher Resilience: The Role of Students and Their Families

Students and their families are considered promoters of resilient behavior in the teacher, where the importance of interpersonal relationships and the socioemotional environment in the educational process is highlighted. In this sense, well-being in the classroom encompasses three characteristics: the student’s emotional management, the feeling of safety provided by the classroom, and the reciprocal trust between student and teacher. The teachers in this study noted that emotionally healthy students, i.e., those who can manage emotions like stress, frustration, and anxiety, allow teachers to remain calm and clear in difficult situations ([59]). This helps teachers make informed decisions and avoid burnout, thus creating a safe and pedagogically motivating classroom environment ([39]; [55]). It should be noted that well-being in the classroom is a reciprocal work performed by the student and the teacher ([68]), where the development of the teacher’s resilience directly impacts the student’s microsystem ([34]; [39]; [42]).

The parent–teacher alliance, established on the foundation of support, commitment, and trust in teachers, builds a connection through dialogue and communication ([24]; [60]; [87]; [90]). Families that show active support and maintain open communication with teachers encourage a collaborative environment that reinforces the development of teacher resilience, where this support manifests as co-responsibility in the education of students ([60]), such as understanding the teacher’s difficulties, cooperation in educational strategies, and reinforcement of values at home that coincide with those taught at school ([55]). When supported by the family, the teacher experiences increased motivation, confidence, and a sense of self-efficacy, creating an environment of collaboration and mutual respect ([87]) and reducing the stress and emotional toll associated with teaching. Knowing that families understand and value their work provides teachers with an additional support network, which promotes their resilience to challenges and helps them face difficulties with a more positive and balanced attitude ([31]; [32]).

The findings in this study suggest that teacher resilience is not a static attribute but a dynamic process that draws from various sources. This study underscores the need for comprehensive approaches that consider personal, professional, and contextual factors in formulating interventions designed to strengthen the qualities that promote resilience in teachers. Elements such as self-care, psychosocial support, and a proactive institutional environment are highlighted as fundamental for teachers to manage the stress and adversities inherent in their work. Furthermore, resilience not only benefits teachers but also has a positive impact on the well-being of students and the educational community as a whole. A resilient teacher is a role model, fostering a positive and motivating learning environment and strengthening interpersonal relationships in the classroom. These findings emphasize the need to implement education policies prioritizing teacher resilience and promoting work environments that favor collaboration and mutual support.

This study has some limitations. The first is the restriction of the sample to elementary school teachers in Chile, so the results are based on the experiences at this level and may differ from those of secondary education or other educational settings. Future studies could extend to private education and different cultural contexts, where the needs and experiences of teachers are as different as they are diverse. Another relevant limitation was the lack of clarity of the concepts held by the teachers. This study examined teachers’ beliefs and self-perceptions; therefore, the ambiguity surrounding certain concepts of interest was useful for identifying patterns necessary for implementing improvements in the school. Future studies could focus on the uniformity of criteria with ex-ante and ex-post follow-up. Together, these approaches will contribute to a more comprehensive framework for addressing resilience in the educational setting, benefiting teachers, their students, and the broader educational community.

## 5. Conclusions

This study provides a dynamic perspective on teacher resilience, considering the diversity of settings in which teachers operate and the factors that promote its development. Teachers’ self-perception of how their resilience is built is revealed as a multifaceted process nurtured by various personal, professional, and contextual elements. In particular, it has been evidenced that self-care, psychosocial support, and a proactive institutional environment are key components that strengthen teachers’ resilience. Similarly, it has been identified that interaction with students and their families plays a crucial role in this process since an environment of well-being in the classroom and a strong partnership between parents and teachers strengthen the resilience of Chilean teachers, particularly those working in vulnerable settings.

Fostering these traits benefits teachers and positively impacts their students’ learning and well-being. Therefore, teacher resilience must be prioritized in education policies and teacher training, including specific programs promoting emotional self-care, psychosocial support, and continuous professional development. The successful incorporation of these findings into educational practice can greatly enhance the growth of healthier and more effective learning environments. This is accomplished by employing strategies that improve emotional and social well-being for both teachers and students, as well as cultivating an atmosphere of collaboration that values mutual support and fosters resilient behavior to deal with academic and personal problems. Thus, both teachers and students can thrive.

Finally, the study emphasizes the value of implementing training programs that emphasize teacher resilience, equipping teachers with resources and strategies to manage stress and adversity more successfully. This would improve teaching quality and cultivate a more positive and collaborative school climate, benefiting the entire educational community. In conclusion, understanding and enriching teacher resilience is essential to sustainable education development in a dynamic environment.

## Figures and Tables

**Figure 1 behavsci-15-00292-f001:**
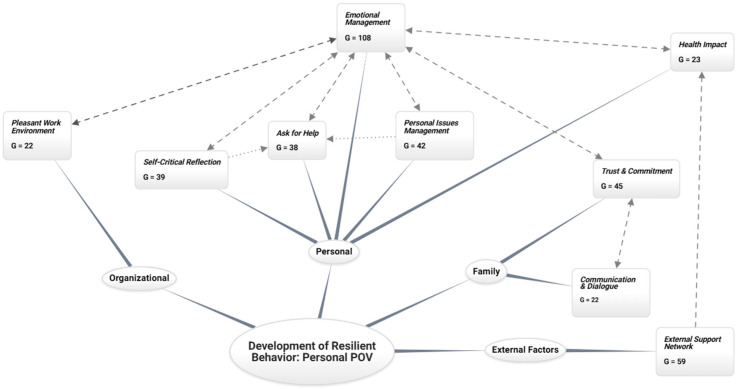
Development of resilient behavior: personal point of view (POV).

**Figure 2 behavsci-15-00292-f002:**
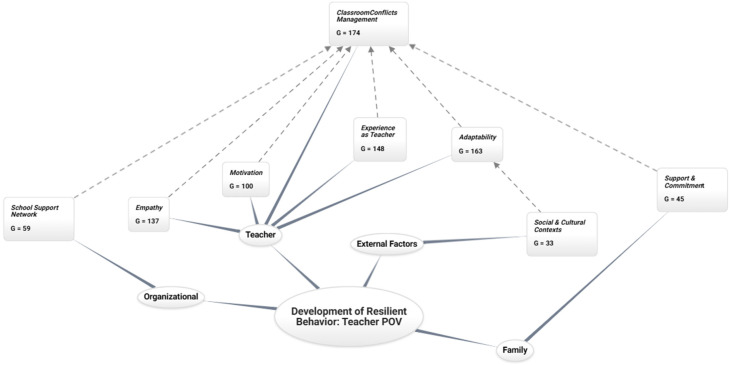
Development of resilient behavior: teacher point of view.

**Figure 3 behavsci-15-00292-f003:**
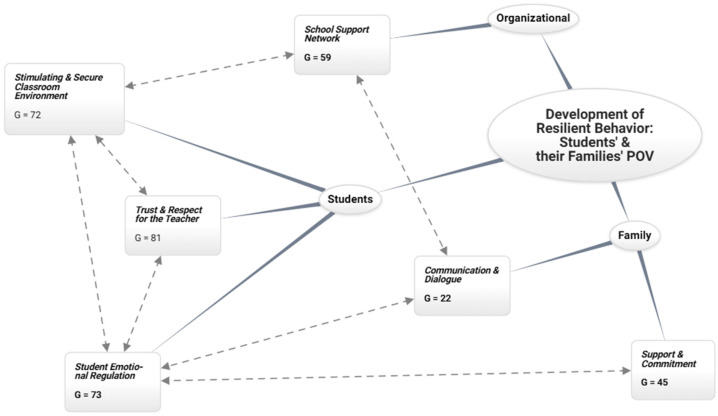
Development of resilient behavior: students’ and their families’ point of view.

**Table 1 behavsci-15-00292-t001:** Demographic and professional characteristics of the participants.

Category	Description
Total participants	63 teachers
Gender	82.5% women, 17.5% men
Average age	40.2 years (SD = 12.3; rank: 24–70)
Years of experience	1–5 years (30%), 6–15 years (45%), more than 15 years (25%)
Educational level	Bachelor’s degree (57%), Postgraduate (43%)

**Table 2 behavsci-15-00292-t002:** General dimensions of resilience development.

Dimension	Description
Students	Students’ perspective coding
Teacher	Teacher’s work perspective coding
Personal	Teacher’s personal perspective coding
Family	Family context coding
External Factors	External factors coding
Organizational	School’s point of view coding

**Table 3 behavsci-15-00292-t003:** Development of resilient behavior: personal point of view: codes.

Personal: *Kindness, Self-Critical Reflection, Self-Care, Adaptability, Dialogue, Positive Feelings, Empathy, Active Listening, Life Experiences, Emotional Management, Personal Issues Management, Health Impact, Ask for Help, Motivation, Optimism, Patience & Tolerance, Persistence & Perseverance, Positive Outlook, Sense of Humor, Personal Values*	Family: *Communication & Dialogue, Trust & Commitment, Responsibility* **External Factors:** *Social & Cultural Contexts, Pandemic, Support Network* Organizational: *Pleasant Work Environment, School Support Network, School Commitment, Educational Modality*

**Table 4 behavsci-15-00292-t004:** Self-perceived teacher’s resilience subdimension: personal categories.

Teacher: *Acknowledging Mistakes, Adaptability, Communication, Peer Trust and Support, Emerging Demands, Dialogue, Empathy, Active Listening, Understanding Student Code, Set Boundaries, Experience as Teacher, Feedback, Building Relationships, Networking, Classroom Conflicts Management, Addressing Student Backgrounds, Motivation, Willpower* Family: *Communication & Dialogue, Support & Commitment, Parent Responsibility*	External Factors: *Social & Cultural Contexts, Pandemic, External Support Network* Organizational: *Pleasant Work Environment, Commitment & Recognition, School Support Network, Educational Modality, School Network*

## Data Availability

The data that support the findings of this study are not available because they are confidential data.

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
