# Peer review of "Understanding Teacher Resilience: Keys to Well-Being and Performance in Chilean Elementary Education"

_behavsci, 2025, doi:10.3390/bs15030292_

Round 1
Reviewer 1 Report
Comments and Suggestions for Authors
I would like to congratulate the authors for selecting a topic that is relevant and timely. They have used a wide range of resources to strengthen their argument, which is commendable. The findings of the paper clearly present the some of the key dimensions of teacher resilience, a contribution that this manuscript makes to the existing body of knowledge. However, there are some areas that need attention. Please see my comments and suggestions.
Introduction:
- Commencing the section with teacher resilience, and its role in the lives of teachers as they navigate various challenges, would be more impactful than introducing the teaching profession and the associated challenges.
- Also, it is unclear what ‘classroom diversity and managing emotions’ mean (lines 38-40). It would be useful to avoid phrases such as ‘strong teaching skills’ (line 43), ‘different fields (line 49) as they seem ambiguous.
- It would be useful to clarify the focus of paragraph 2 (lines 48-63).
- In the second paragraph, the authors mention that resilience leverages on different levels that include personal, social and contextual. Lines 56-59 give the impression that these are the major focus points, however, these are not explored in detail. Offering a definition can help, especially if these offer a framework for the subsequent paragraphs.
- In the third paragraph, the authors mention socio-emotional competencies. It is important to clarify what they mean in the context of resilience. Such concepts need sufficient detail. Additionally, there is a sudden jump to teachers with low and high resilience without much explanation.
- There are quite a few things addressed between lines 64 and 121 – for example, social and emotional competencies (lines 64-81), professional environment (lines 82-98), teachers’ families (lines 99-106), students’ families (lines 107-121). It would be useful to clarify if these are related to personal, social and contextual factors.
- It is critical to include empirical data that tells us about resilience in socio-cultural contexts like Chile. While the citations are useful, it would be useful to indicate whether these are from elementary school teachers’ perspectives.
- Also, authors should include some contextual information about Chile that has prompted this research.
Methods
- I suggest presenting participants’ info in a table for better clarity.
- It would be useful to include some information on the participants’ years of work experience.
- Elaborating on lines 173-175 for better clarity will be useful.
- The data is gathered from elementary school teachers in Chile, therefore it is essential to provide some contextual information, so that the readers get a better picture of teacher resilience as perceived by the Chilean educators.
Findings
I particularly appreciate the findings section, which is well-supported by data. However, the results present a somewhat generic overview of all participants, despite some diversity in the participant profiles (as presented in the methods sections). It would also be beneficial to include some analysis based on the participants’ number of years of work experience in the sector, as this could provide a more nuanced perspective on resilience, which the paper could explore further.
Discussion:
To enhance the paper, it would be beneficial to include additional insights on resilience within the socio-cultural context of Chile. Furthermore, incorporating a discussion that relates to existing empirical evidence will add valuable depth. It would be useful to revisit sections 4.1 and 4.2 as there are considerable overlaps between them.
Comments on the Quality of English LanguageThe paper needs close editing for coherence, transition between paragraphs, and between sentences within paragraphs.
Author Response
Dear revisor,
Thank you for your comments, which have helped us improve the manuscript. Various enhancements have been made to the article, highlighted in yellow.

Reviewer 2 Report
Comments and Suggestions for Authors
Understanding teacher resilience: keys to well-being and performance in Chilean elementary education
The study touches upon a fundamental issue. Therefore, I congratulate the researchers. The up-to-date nature of the resources used in the study is noteworthy. As a contribution to this qualified study, I would like to offer some suggestions to researchers.
Lines 30-31: keywords must be relevant to your research. Appropriate keywords will increase the visibility of your research when searched. For example, words such as well-being, and resilience can be included.
Line 49: Would it be better to use profession instead of job?
Line 56: It may be helpful to explain the topic's connection to teacher training.
There are many studies in the literature on the well-being of teachers. Many aspects of the subject have already been addressed and examined. Therefore, it would be useful to emphasize the possible benefits of the subject for education in your own country a little more clearly. This will allow us to better understand the difference and importance of your research. What kind of education system is there in your own country, what is teacher training like, what are the working conditions of teachers? What are the salaries, workload, working environments and opportunities offered to teachers? What kind of results have previous studies on this subject in your country produced? The subject can be addressed within the framework of these. When we know all these, the research results will become more clear.
What are the types of resilience? It would be more enlightening to reveal the relationship between resilience and well-being more specifically. One cannot be accepted as the absolute result of the other.
lines 151-153: this was a difficult sentence for me to understand.
Line 160-175: Teachers' years of experience and the student population they work with may also have an impact on well-being. The majority of participants appear to be female. What results have studies in the literature revealed regarding gender or experience? There is no information on this. More information about the participants would be helpful.
Line 169: Including information such as how the questions used in the research were developed and whether expert opinions were obtained will make the research method more effective. What method was followed for reliability?
Lines 199- 465:The findings are presented in a very clear and understandable manner. Thank you.
Line 602: The conclusion should be reviewed in the light of the available data and should be expressed more clearly (in parallel with the findings). I believe that establishing a connection between the discussion and the introduction will facilitate the integration of the subject in the reader's mind. In addition to these, the suggestions to be presented by the researchers can also be guides for subsequent researchers.
In brief, I think that the results and discussion sections are presented in a very high-quality manner. However, I think that the introduction and conclusion sections can be improved. It would be useful to make some additions to the “participants and data collection tools” section as well.
Author Response

(The authors gave the same response as above.)
